# Optimization of Apex Shape for Mounting to the Bead Bundle Using FEM

**DOI:** 10.3390/ma16010377

**Published:** 2022-12-30

**Authors:** Peter Palička, Róbert Huňady, Martin Hagara, Pavol Lengvarský

**Affiliations:** Department of Applied Mechanics and Mechanical Engineering, Faculty of Mechanical Engineering, Technical University of Kosice, 04001 Kosice, Slovakia

**Keywords:** pneumatic tire, optimization, finite element method, apex

## Abstract

Tires are one of the most basic and important components of vehicles, including bicycles, cars, trucks, and aircraft. They consist of several layers that provide complex and dynamically changing functions. This work aims to optimize the mounting process of the tire apex to the bead. The bead locks the tire to the rim and helps minimize the risk of rim slip, and the apex provides dynamic stiffness, stress distribution, and driving stability. In mounting the apex onto the bead, air can be trapped between the apex and bead, which is an undesirable and significant problem in tire manufacturing. An FE model was created to simulate and optimize this process. After modifying the apex dimensions, the air was displaced from the space between the apex and the bead. Based on the simulation results, a set of recommendations for producing suitable apex shapes is provided.

## 1. Introduction

The modern radial tire is composed of various materials, chemicals, fabrics, and steel reinforcements, making it one of the most complex composites in mass production. This is due to the complexity of the material blends or compounds used in the tire, the quantity and variety of compound types, the complexity of the fabrics and steel reinforcements, and the high level of uniformity and consistency required to make the tire work on today's vehicles. Regardless of tire design or use, all tires must meet essential functions, such as providing load-carrying capacity, cushioning and damping, transmitting driving and braking torque, generating steering response, providing minimum noise and vibration, etc. The modern tire in today's global market falls into two types: bias-ply construction and radial [1].

Tire fabrication is a complex process consisting of many steps that can lead to errors associated with the manufacturing technology. One of the errors that can occur is that the apex of the tire can trap air when mounted onto the bead and separator. Trapped air is a significant problem in tire manufacturing. Therefore, this process needs to be optimized by parameterizing the apex or changing the separator’s radius.

Optimization of rubber properties, shape, and design is a task for the manufacturing process. In recent years, rubber isostatic pressing (RIP) has been developed [2,3,4,5]. Shima et al. [6] studied numerically (using FEM) and experimentally fundamental rubber powder isostatic pressing features to perform a net-shape process. It turns out that the resulting shape of the compact depends on the material properties of the rubber and the thickness of the rubber mold. The shape of the mold can be optimized so that the mold provides the required compact shape with the desired mean relative density. When determining the shape of the mold cavity, the deformation of the compact caused by sintering must also be taken into account.

Shape optimization of rubber components [7,8] in the automotive industry is a commonly treated issue in designing the shape or design of these products and maintaining their functional properties. Kim et al. [9] optimized the mounting performance of a grommet using EPDM materials. Ethylene propylene diene monomer (EPDM) has excellent mechanical properties and is therefore used in many products. In the automotive industry, this material is used to make grommets, which are an essential part of cars. The authors have experimentally investigated the physical properties of the primary molding materials as a function of process parameters. The grommet was fabricated in accordance with the manufacturing process parameters, and the insertion and separation forces were investigated. 

Other important tire characteristics that receive much attention in practice are driving behavior, noise, and comfort. Tire noise is closely related to driving comfort. It is preferable if the tire has a low noise level; therefore, it is necessary to evaluate how the rolling noise level varies depending on the type of tire, road surface, vehicle, or driving mode [10]. The tire is in contact with different roadways, where potholes and various road damage can be found. Therefore, a significant part of research papers focuses on the formation or tendency of the tire to resist cracks [11] and the interaction between asphalt roadways and pneumatic tires [12,13]. Research papers are also devoted to studying tire rolling deformation [14] and investigating the hyperelastic properties of retreaded tire rubber using the finite element method [15].

Researchers often aim to improve the tire manufacturing process or various driving characteristics of tires using optimization techniques and ANNs (artificial neural networks) [16]. Wang et al. [17] proposed a reasonable prediction method to determine the fatigue life of radial tire bead. The damage that occurs around the tire bead region is one of the critical failure forms of a tire. Generally, the prediction of tire durability is performed by the experimental method. However, conducting experiments requires a lot of money and time. Fatigue testing of bead rubber compounds was used to determine the range of strain energy density. The maximum strain energy density range of several bead compounds was obtained by steady-state rolling analysis with the finite element method. This quantity was then inserted into a fatigue life equation to estimate the fatigue life. This method can be effectively used to predict the fatigue life of a tire bead as the experimental results were in good correlation with the estimated value. It has been shown that the carcass plays a crucial role in the fatigue life of the tire bead, and targeted optimization of this compound may effectively improve the fatigue performance of the tire bead.

Tire research in terms of tire shape and design, tire production, and the optimization of the manufacturing process is a regular topic of research [18,19,20,21]. Liu et al. [22] conducted a study to support the development of run-flat tires. They investigated two key factors that affect the thermal performance of the rubber insert and the stress distribution on the tire sidewall. These key factors of heat effect and stress concentration were analyzed in detail through various performance tests and simulations. A two-dimensional model of the tire was used for the simulations.

Li et al. [23] analyzed the development trend of the tire bead and proposed its new topological structure for it. In the research, they proposed a complete set of manufacturing technologies, from the processing of the whole tire bead to the validation of the tire, which was well implemented. In their work, they used experimental and numerical methods of mechanics.

This study aims to create a finite element model (FE model) simulating the mounting of the apex onto the bead and to optimize this process. The FE model is created and simulated in Abaqus and optimized in Isight software. Once the simulation results had been achieved, several approaches to optimize the process were tested. The method that demonstrated the best results was selected and is presented in this paper. This study is unique because most articles and publications deal with the driving characteristics of the tires and their behavior on the road or in different terrains [24,25,26,27,28,29], not with the analysis of the tire manufacturing process. Finite element analysis (FEA) of tires has become an essential part of the design cycle for most tire manufacturers. It is widely used to predict the behavior of a virtual tire under different real conditions, ranging from simple loads such as inflation to highly complex situations such as transient moving, non-isothermal steady-state rolling, and material loss during wear on rough surfaces. Developing sophisticated adaptive finite element techniques has recently increased researchers' interest in tire-wear modeling.

## 2. Apexing on the Apexing Machine

The role of the tire is to ensure immediate contact with the road. It has to carry the vehicle load and provide torque transmission and steering response to ensure satisfactory driving performance. The bead serves to lock the tire to the rim and helps to minimize the risk of rim slip, i.e., circumferential tire slip on the rim, due to occasions of high drivetrain torque. The bead (Figure 1) is produced by spiral winding of brass-coated or sometimes bronze-coated steel wire, which is previously coated with a compound called “bead insulation”, to form a hoop of the specific dimension of the tire. The apex is the component above the bead, filling the space between the inside ply and the ply turn-up. It is usually a hard compound, which contributes to the bead stability but also helps to provide a modulus gradient up the sidewall, preventing stresses from developing towards the high flexural zone at the center of the sidewall [1,30]. 

Apex is mounted to the bead bundle using an apexing machine. There are several apexing machines available on the world market. Well known is the VMI APEXER featuring VMI’s extrusion technology that automatically assembles freshly extruded apex with a pre-manufactured bead, ensuring a consistent high-quality bead apex. VMI machines are designed for short apexes. If apexes are longer, then a dual apexing machine (DAM) is used because its design is more suitable. The VMI APEXER operates with short apexes, so it does not need pre-shaped separators. However, the separator has to be used since DAM is designed for long apexes that can bend and stick together. The separator is a plastic ring. It is performed so that the apex is bent more flexibly on the tire-building machine. It is necessary to have a pre-shaped apex to bend it properly when the tire is built and not to occur production faults through trapped air. Trapped air is a significant problem in the manufacturing of tires. The principle of the DAM machine is pneumatic positioning of the apex on the bead using an inflatable membrane (bladder) that can work with two cores simultaneously. During the inflation and deformation of the bladder, the apex is lifted and inverted and then fitted on the bead, see Figure 2 and Figure 3.

## 3. Yeoh Hyperelastic Material Model

Apex is made of rubber, which is a hyperelastic material. Since such a material exhibits an instantaneous elastic response up to large deformations, it cannot be described by material constants such as Young's modulus and Poisson's ratio, as in the case of steel. A suitable theory has to be chosen to describe the deformation of a hyperelastic material. In this work, the Yeoh hyperelastic material model is used. It is an isotropic and nonlinear material model. The form of the Yeoh strain energy potential is [31]
(1)$$U=C_{10}({\bar{I}}_{1}-3)+C_{20}{\left({\bar{I}}_{1}-3\right)}^{2}+C_{30}{\left({\bar{I}}_{1}-3\right)}^{3}+\frac{1}{D_{1}}{\left(J^{el}-1\right)}^{2}+\frac{1}{D_{2}}{\left(J^{el}-1\right)}^{4}+\frac{1}{D_{3}}{\left(J^{el}-1\right)}^{6},$$
where U is strain energy per unit of reference volume; $C_{i0}$ and $D_{i}$ are temperature-dependent material parameters; and ${\bar{I}}_{1}$ is the first deviatoric strain invariant defined as [31]
(2)$${\bar{I}}_{1}={\bar{\lambda }}_{1}^{2}+{\bar{\lambda }}_{2}^{2}+{\bar{\lambda }}_{3}^{2},$$
where the deviatoric stretches ${\bar{\lambda }}_{i}=J^{-\frac{1}{3}}\lambda_{i}$; J is the total volume ratio, $J^{el}$ is the elastic volume ratio, and $\lambda_{i}$ are the principal stretches. The initial shear modulus and bulk modulus are obtained by [31]
(3)$$\mu_{0}=2C_{10},K_{0}=\frac{2}{D_{1}}.$$

## 4. Finite Element Model and Simulation

The model consists of four parts: apex, 17-inch bead, bladder, and separator. Since it is an axisymmetric problem, it can be solved using a two-dimensional model (see Figure 4) that assumes rotational symmetry in geometry, material properties, boundary conditions, and loads. It can lend itself to a cylindrical coordinate. The dimensions of the apex before optimization are shown in Figure 5.

The bladder and separator are modeled as analytical rigid. This is because it is not necessary to determine stress and strain values in these parts of the model, and they are much stiffer than the parts relevant to the analysis. Therefore, no material properties are assigned to them and there is no need to mesh them either, which saves computational time and memory. The bladder geometry is simplified for this simulation task. The real bladder is a membrane that inflates and pushes the apex onto the separator and the bead (see Figure 3). The bladder moves the apex to the separator. When the apex touches the bead of the tire, the apex loses contact with the bladder and strikes the separator. In the real manufacturing process, the separator is plastic, but since the apex is made of rubber with a stiffness much less than the stiffness of the separator, it can be modeled as rigid. It reduces the complexity of FE model building and also the simulation time.

In the next step, material properties were assigned to the apex and bead of the tire. While the apex is made of rubber only, the bead is a composite that consists of a rubber filler and steel wires. A Yeoh hyperelastic material model was used to define the rubber. Steel was defined as a linear elastic material. The material constants used are shown in Table 1. 

The apexing process was simulated using two transient quasi-static Visco steps: apexing and pressing. The interactions between the model parts were defined as surface-to-surface using contact pairs. For all pairs, a "hard contact" was set, and a penalty method was used. The other interaction properties for the tangential and normal behavior of the contacts are provided in Table 2. The penalty method was used to describe the nonlinear behavior in the normal direction. The contact stiffness parameters used are listed in Table 3.

The next step was to define the boundary conditions and loads. In the apexing step, the bladder moves the apex in the x-axis direction by 17.2 mm while rotating about the z-axis by 1.6 radians (91.673°). These movements were defined as forced displacements. While the displacement U1 increases linearly with time from the beginning of the step, the rotation UR3 of the bladder is zero within 0.3 s. Therefore, the amplitude of this rotation had to be entered in tabular form (see Table 4). The remaining degrees of freedom, U2 and UR2, were removed. Since the bladder is an analytical rigid body, the above boundary conditions were defined at its reference point. The bead can only move in the x-axis direction due to the apex mounting on the separator (apexing step). Thus, displacement U2 in the y-axes’ direction and rotations UR2 and UR3 about y- and z-axes, respectively, were set to zero. The separator is fixed, so all degrees of freedom are removed at its reference point. Since the apex moves freely in both computational steps, it was necessary to activate the automatic stabilization of the model. In the pressing step, a normal pressure with 0.01 MPa amplitude is applied to the apex to press it onto the bead and separator. For the present analysis, the state after apex unloading is crucial. Therefore, the time dependence of the pressure amplitude was also defined in tabular form (see Table 4). The definition of the boundary conditions and loads is shown in Figure 6.

Linear quadrilateral elements of type CGAX4RH were used to mesh the bead and apex. The apex mesh contained 114 elements with an average size of 1.47 mm. In the case of the bead, a finer mesh was used because it contains steel wires for reinforcement, which require smaller elements. The bead mesh contained 438 elements with an average size of 0.39 mm. The resulting FE mesh (Figure 7) consisted of 552 elements and 610 nodes. The analytical rigid parts (bladder and separator) are unmeshed. 

The application of the apex onto the bead, obtained by simulation in Abaqus, is shown in Figure 8. The figure shows the model transitioning from its initial position to the position at the end of the apexing step (marked as intermediate position) and then to the position at the end of the pressing step (marked as final position).

Figure 9 shows the von Mises stress and contact pressure at the end of the simulation when the pressure acting on the apex is removed to allow the apex to release elastically. As can be seen, the air is trapped between the tire's bead and apex, i.e., a manufacturing problem that needs to be eliminated. After a detailed investigation and considering all aspects, it became apparent that the resulting contact pressure on the apex significantly influenced the formation of air traps. 

## 5. Optimization of Apex Shape

The optimization aims to obtain such a shape of the apex that no air traps are formed when it is mounted onto the bead. Based on the conclusions of the numerical analysis, it was decided that the objective function would be to maximize the minimal contact pressure between on the apex and the separator at the end of the simulation. The intention is also to increase the contact area between the apex and the bead. The optimization was performed in Isight software. As can be seen in Figure 10, the sim-flow scheme consists of only two components, optimization and Abaqus. After the optimization task is set, the optimization process starts. The initial model is loaded into Abaqus, and the calculation is performed. Isight suggests new apex dimensions, sends the updated Abaqus file to the solver, and re-simulates the model. The results are returned to Isight, which evaluates whether the solution was feasible, infeasible, or optimal. Each solution is compared with the previous one and evaluated which is better. 

The Pointer–Pointer automatic optimizer was used for optimization because it is a technique that disposes of four optimization methods: genetic algorithm, Nelder and Mead's descent simplex, sequential quadratic programming (NLPQL), and linear solver. This technique can work with only one of them or with all of them at the same time. After running the optimization, the program evaluates which method is the best for solving the problem and defines the most efficient control parameters, step sizes, number of iterations, etc. The objective function was set to maximize the minimum contact pressure value. Essentially, this means that Isight is looking for the loading pressure that causes the maximum pressure on the contact surface of the apex. The proposed optimization variables are introduced into the FE model in each iteration step. Subsequently, a simulation is performed in Abaqus, resulting in the maximum contact pressure value. The optimization data flow is shown in Figure 11. The design variables in the optimization were the apex dimensions characterizing its shape (see Figure 5) and the pressure by which the apex is pressed against the separator represented the output variable. Only dimensions *a*, *b*, *c,* and *d* were optimized. When all dimensions were included, the 0.6 mm and 1 mm dimensions introduced errors into the solution (see Table 5), or there were cases where the design was feasible but the values of the optimized dimensions were negligible.

## 6. Results and Discussion

The optimization algorithm performed 78 iterations within the solved optimization loop, of which 71 are feasible and seven are infeasible. It means that the optimization is set up well, and there are no errors. Out of all the possible solutions, Isight has selected the most optimal one. It is shown in Figure 12.

The von Mises stress and contact pressure of the optimum apex design are shown in Figure 13. As seen, the pressure on the apex and bead contact surface was increased as required and set in the optimization. At the same time, the contact area was increased. This ensures better adhesion between them. Although the apex optimal dimensions are not too different from the original ones, the results indicate that even a minimal change in dimensions results in a better fit of the apex onto the bead and separator. On the other hand, a significant change in the apex dimensions leads to significant manufacturing errors, as shown in Table 5. In a real production, such errors are often evaluated by the trial-and-error method. However, it has been shown that numerical optimization makes it possible to obtain an optimal design with high cost and time savings.

Although the results show a better fit of the apex to the separator and bead, it is also important to note that there are a lot of simplifications in the model. The entire problem is solved using a two-dimensional model that assumes rotational symmetry of geometry, material properties, boundary conditions, and loads. In addition, the bladder and separator were modeled as rigid bodies. Based on the results obtained from this study, however, it is possible to define the following practical recommendations:Do not produce thick apexes. The results have shown that a thickness of 9 mm suits perfectly. Apex with a thickness of, e.g., 11 mm causes large deformations in the beam.Do not produce apexes thinner than 8 mm as they do not fit the separator over the entire contact surface.Do not change the shape and dimensions of the apex tail. The optimal length of the tail is 6.5 mm.

It is important to note that these recommendations are valid for the bead of the given shape and dimensions.

When designing the geometry of an apex, the tire engineer is interested in its performance characteristics to design a tire with the desired properties (handling, rolling resistance, etc.). A design conflict will occur if the apex is difficult to manufacture or mount correctly on the bead. The final performance requirements are often in conflict with the requirements of the tire manufacturing process. Only manufacturing requirements are considered in this study.

## 7. Conclusions

This study aimed to create a FE model simulating the mounting of an apex onto a plastic separator and to optimize this process to eliminate air traps. For this purpose, a simplified axisymmetric model was created, which consisted of four parts. Two of them, the separator and the bladder, were modeled as analytically rigid bodies. The apex and bead were deformable. The apexing process was simulated in Abaqus through two Visco-type steps. The optimization was performed in Isight, one of the most powerful tools. Before the final optimization, four approaches to optimize the apexing process were tested. The original idea was to optimize the dimensions of the apex and the separator's radius, with the objective function maximizing the minimum pressure value at the contact surface of the apex and the separator. An alternative was a multicriteria optimization with two objective functions, where the maximum value of contact pressure was minimized, and the minimum value of contact pressure was maximized to achieve its better distribution. The third option was to optimize only the apex dimensions and maximize the pressure. These approaches did not provide an optimal solution. Indeed, for a real process, it is more efficient to modify the shape of the apex than the shape of the separator since the change of the separator implies additional production costs. The most suitable option appeared to be optimization of the apex where the minimum value of the contact pressure between apex and separator was maximized. The obtained results showed a better fit of the apex to both the separator and the bead bundle, thus eliminating the air-trapping problem.

## Figures and Tables

**Figure 1 materials-16-00377-f001:**
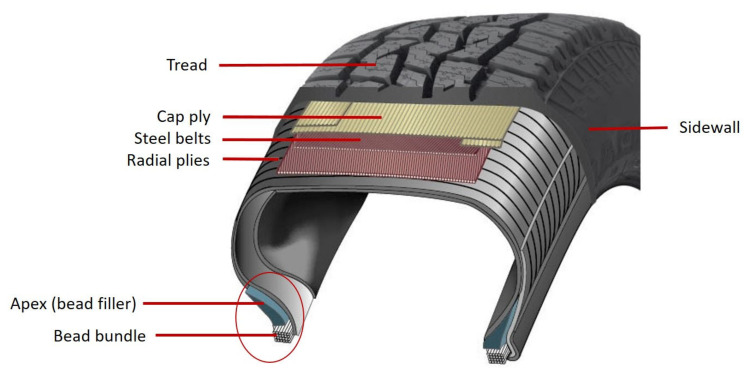
Composition of a tire.

**Figure 2 materials-16-00377-f002:**
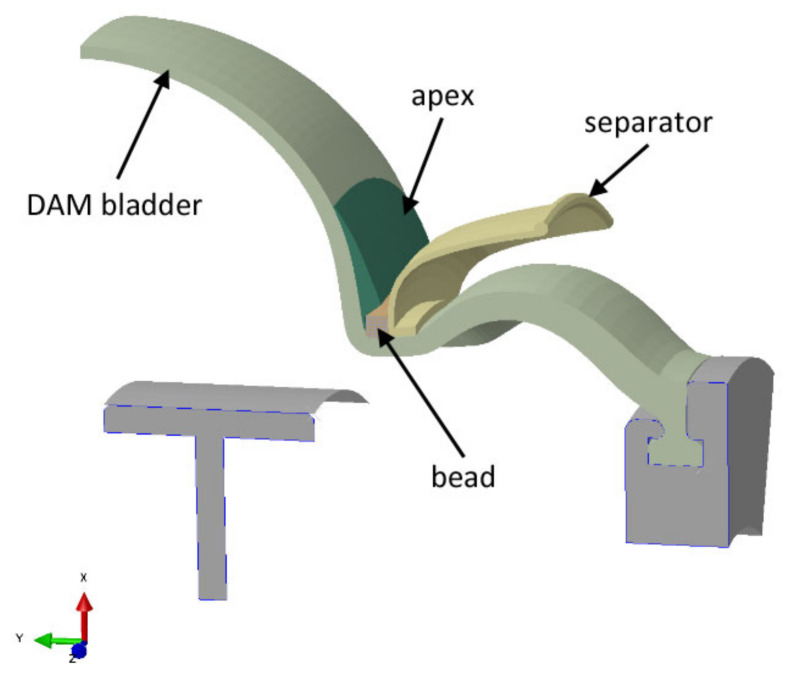
Mounting the apex on the bead using an inflated bladder on a DAM machine (symmetrical half view).

**Figure 3 materials-16-00377-f003:**
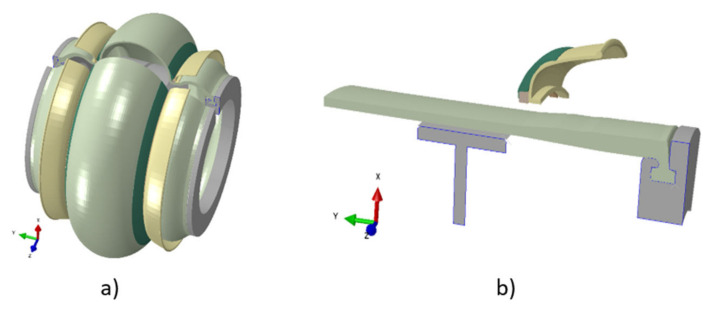
(**a**) Overall view of apexing using a DAM machine; (**b**) apex mounted on a bead, supported by a separator.

**Figure 4 materials-16-00377-f004:**
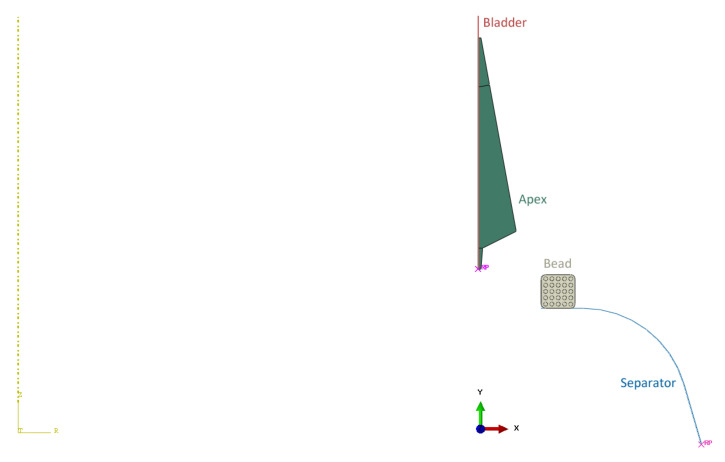
Model assembly.

**Figure 5 materials-16-00377-f005:**
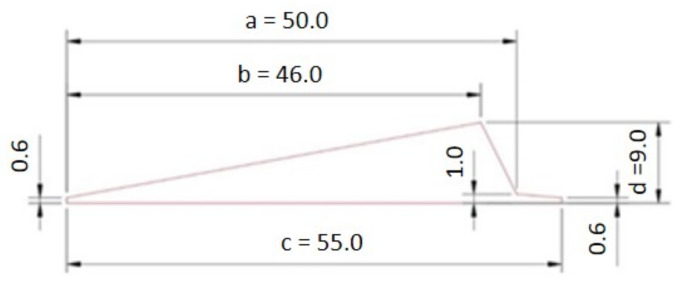
Dimensions of apex before optimization.

**Figure 6 materials-16-00377-f006:**
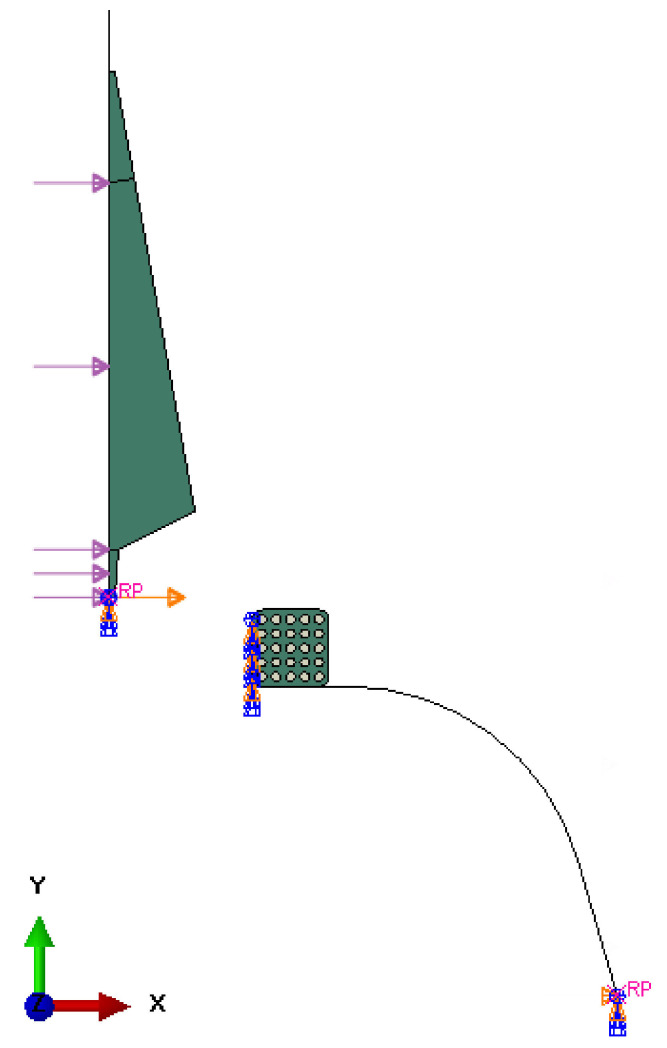
Boundary conditions and loads.

**Figure 7 materials-16-00377-f007:**
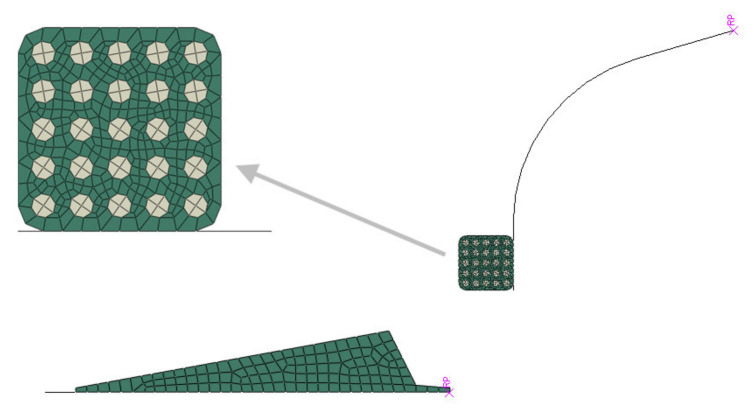
FE mesh of the model.

**Figure 8 materials-16-00377-f008:**
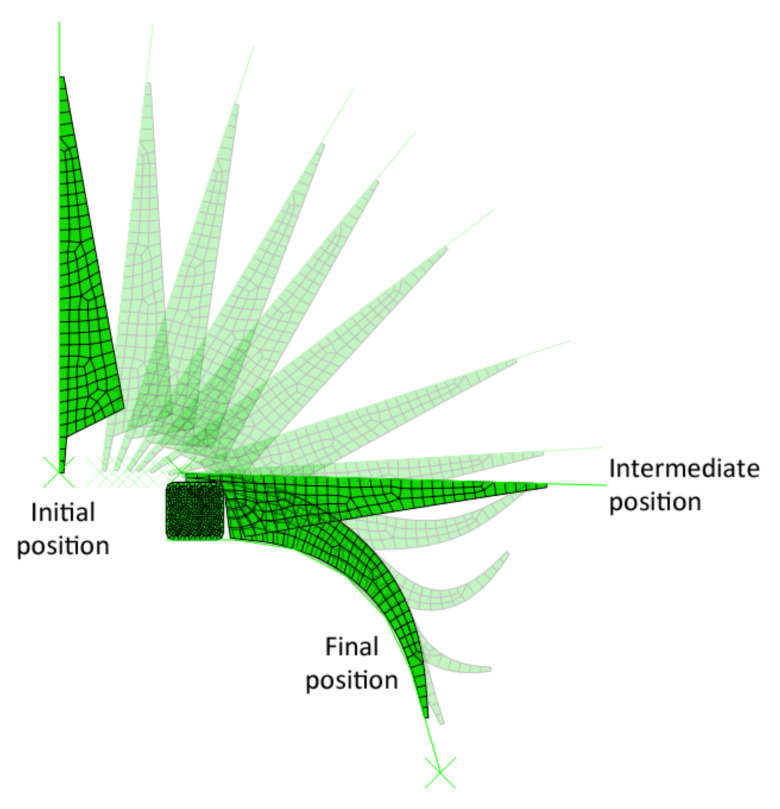
Simulation of the apexing process.

**Figure 9 materials-16-00377-f009:**
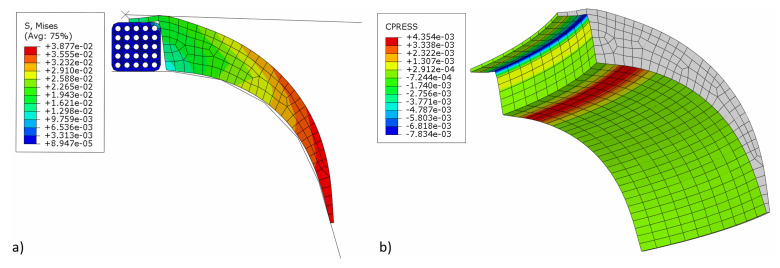
Simulation results before optimization: (**a**) von Mises stress (MPa); (**b**) contact pressure on the apex (MPa).

**Figure 10 materials-16-00377-f010:**
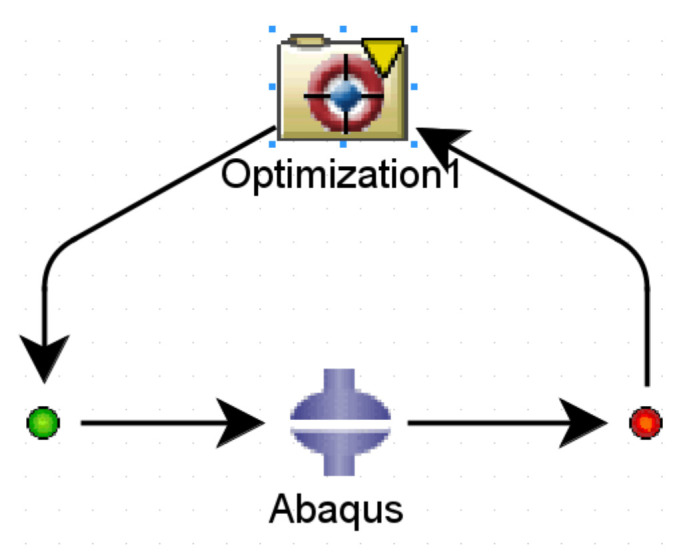
Optimization sim-flow.

**Figure 11 materials-16-00377-f011:**
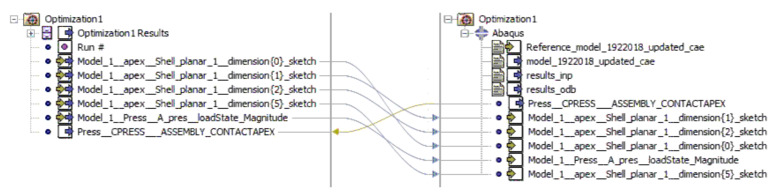
Optimization dataflow.

**Figure 12 materials-16-00377-f012:**
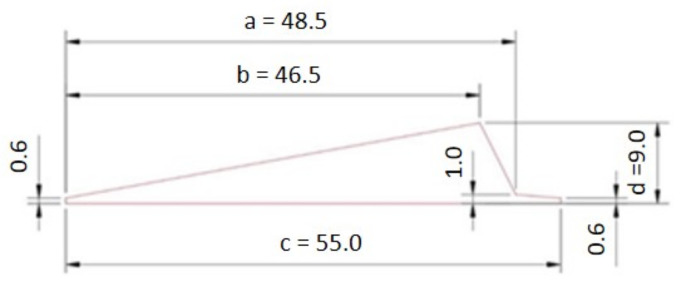
The resulting shape and dimensions of the apex after optimization.

**Figure 13 materials-16-00377-f013:**
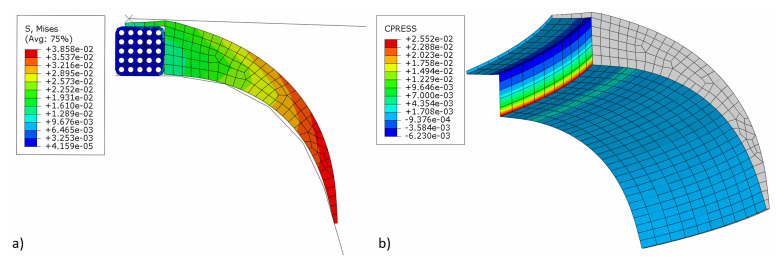
Simulation results after optimization: (**a**) von Mises stress (MPa); (**b**) contact pressure on the apex (MPa).

**Table 1 materials-16-00377-t001:** Material data.

Steel	Rubber
Poisson’s ratio: 0.3	Yeoh’s coefficients (MPa):
Young’s modulus: 200.000 MPa	*C*_10_ = 0.025, *C*_20_ = 0, C_30_ = 0.002 *D*_1_ = 0.001, *D*_2_ = 0, *D*_3_ = 0

**Table 2 materials-16-00377-t002:** Interaction properties.

		Tangential Behavior		Normal Behavior
Contact Pair (Master/Slave)	Step	Friction Formulation	Friction Coefficient	Separation
Bladder/Apex Separator/Bead	Both	Penalty	1.0	Allowed
Apex/Bead	Apexing	Penalty	1.0	Allowed
	Pressing	Rough	-	Not allowed
Separator/Apex	Pressing	Penalty	1.0	Allowed

**Table 3 materials-16-00377-t003:** Contact stiffness parameters.

Normal Behavior	Nonlinear
**Stiffness scale factor**	0.1
**Initial/Final stiffness ratio**	0.01
**Upper quadratic limit scale factor**	0.03
**Lower quadratic limit ratio**	0.33333
**Clearance at which contact pressure is zero**	0

**Table 4 materials-16-00377-t004:** Amplitude data.

	Rotation UR3 of the Bladder (Apexing Step)		Pressure on the Apex (Pressing Step)	
	Time (s)	Amplitude (rad)	Time (s)	Amplitude (MPa)
1	0	0	0	0
2	0.3	0	0.5	0.01
3	1	−1.6	1	0

**Table 5 materials-16-00377-t005:** Not feasible designs from optimization.

Design of the Apex	FEM Simulation Results
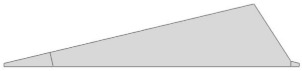	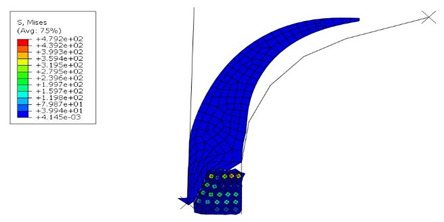
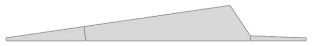	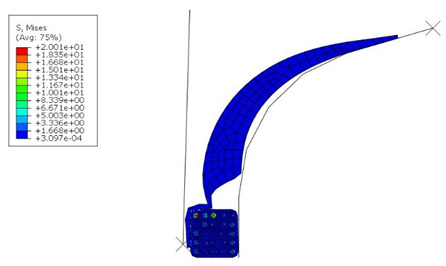
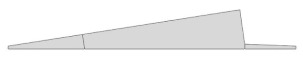	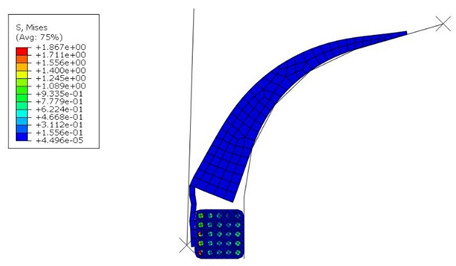
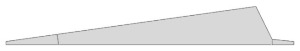	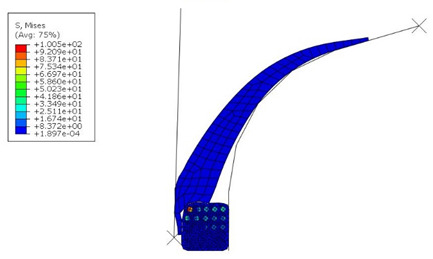

## Data Availability

Not applicable.

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
