# Peer review of "Optimization of Apex Shape for Mounting to the Bead Bundle Using FEM"

_materials, 2022, doi:10.3390/ma16010377_

Round 1

Reviewer 1 Report

It is a very well presented study. It is quite simple, but it seems of importance for a specific branch of industry. It presents very clear a problem, its treatment and solution. The treatment given for the materials involve seem adequate.

Table 2, could be reduced if they take of columns of pressure-over and constraint enforcement

Author Response

Dear reviewer,

We appreciate the time and effort you have dedicated to providing valuable feedback on our manuscript. We are grateful for your comments and recommendations that improve the level of our paper. We have successfully incorporated all your suggestions. All changes in the revised manuscript, including changes based on other reviews, are highlighted.

Kind regards,

Authors

Response to Reviewer 1 Comments

  1. Table 2, could be reduced if they take of columns of pressure-over and constraint enforcement.
    • We have reduced the table and inserted the excluded information into the text on page 7.

Reviewer 2 Report

Major comments:

This paper provides a numerical study on mounting an apex onto a plastic separator via the FEM, where an optimization process is added to eliminate the air traps. However, the results-and-discussion section fails to provide a detailed discussion to convince the reviewer that their FE model works correctly. It is more like a technical report rather than a scientific paper. I suggest the authors to enrich the literature review of this field, and specify their research topic into a narrower scenario, from where a crucial science problem may be found. I believe the presented method is a good numerical approach to solve such issues.

Additional:

Table 1: please provide the data source reference of your material properties

Equations 1 and 2 need references (can be textbooks..), just like Equation 3

Mesh sensitivity of the FE model should be clarified, where a study on the computational accuracy of a coarse, fair, and fine meshes is needed.

Figure 11 needs a more detailed explanation, which can provide readers a more clear view of how the optimization was performed

Figure 12: I suggest the decimal digits of all dimensions stay the same, e.g., 0.6, 9.0, 55.0 …

Is there a way to convince readers that your FE results and optimized results are correct? Maybe a similar computational study’s results are in line with yours, or there are comparable experimental results that support your results. Please provide.

The practical suggestions provided starting from Line 301 are useful, I believe, while are too general. Please provide more quantitative data to limit the applicability of your suggestions, this also makes your suggestion more useful in the special scenario that you had limited.

Author Response

Dear reviewer,

We appreciate the time and effort you have dedicated to providing valuable feedback on our manuscript. We are grateful for your comments and recommendations that improve the level of our paper. We have incorporated changes to reflect most of your suggestions. Please consider that some of the other reviewers' suggestions conflicted with yours. All changes in the revised manuscript are highlighted. A point-by-point response to your comments and recommendations can be found in the attached PDF file.

Kind regards,

Authors

Reviewer 3 Report

The manuscript presents simulation FEM results aiming to produce suitable apex shapes via optimization. It is well written and is interesting. However some points need to be addressed and revised:

-In the Introduction section the paragraph the third paragraph that describes a summary of the work and the novelty of the work should be at the end of the Introduction.

-The part of the Introduction where literature work is described should be reduced in size and to keep the parts that are most relevant to the manuscript. A thorough description of the different litterature works should be avoided.

-What type of finite element analysis was conducted in Abaqus? Transient or static? More details should be given

-The total number of elements for the simulation for all 4 parts should be mentioned.

-Figure 11 that demonstrates the optimization data flow should be analyzed and more details should be provided.

-Which optimization method produced better results? This should be discussed on the results section not only in the conclusions. And more details should be given for this optimization method and why it was more efficient.

-The authors claim that "it is also important to note that there are a lot of simplifications in the model". These simplifications should be mentioned in this part of the manuscript.

-The following part in the conclusion section

"The original idea was to optimize the dimensions of the apex and the separator's radius, with the objective function maximizing the minimum pressure value at the contact surface of the apex and the separator. An alternative was a multicriteria optimization with two objective functions, where the maximum value of contact pressure was minimized, and the minimum value of contact pressure was maximized to achieve its better distribution. The third option was to optimize only the apex dimensions and maximize the pressure. "

is not addressed with a lot of details before the conclusion section. This should be done more properly in the previous sections since it is an important point for the manuscript.

Author Response

Dear reviewer,

We appreciate the time and effort you have dedicated to providing valuable feedback on our manuscript. We are grateful for your comments and recommendations that improve the level of our paper. We have incorporated changes to reflect most of your suggestions. Please consider that some of the other reviewers' suggestions conflicted with yours. All changes in the revised manuscript are highlighted. A point-by-point response to your comments and recommendations can be found in the attached PDF file.

Authors

Round 2

Reviewer 2 Report

I want to thank the authors for their efforts on addressing my concerns. Nevertheless, some of the issues are not resolved well: e.g., response of comment #7 said that their numerical results were only validated verbally by the customer. I do not believe this is what a scientific paper should look like. To this end, I recommand a rejection.

Author Response

Dear Reviewer,

we respect your opinion, but we are unable to accommodate your request. The subject of our paper is an optimization calculation aimed at removing air traps in apex fitting. This is a specific problem that is not solved in the literature. The question of experimental verification of the solution proposed by us is possible, but it is not within our capabilities, since we do not have access to the necessary technological equipment of the manufacturer, which is, moreover, a protected production secret. The aim of the paper is to present a methodology for solving the problem in the process of development and production of tires.

We will be glad if you reconsider your decision. 

Kind regards,

Authors

Reviewer 3 Report

The manuscript has been improved.

As a last comment I suggest you add the corresponding references in the following part of the text

"This study is unique because most articles and publications deal with the driving characteristics of the tires and their behavior on the road or in different terrains [ ], not..."

Author Response

Dear Reviewer,

The requested references have been added to the manuscript. Thank you for your review.

Kind regards
Authors